META-RESEARCH ARTICLE

# Creating clear and informative image-based figures for scientific publications

Helena Jambor[1‡], Alberto Antonietti[2,3‡], Bradly Alicea[4], Tracy L. Audisio[5], Susann Auer[6], Vivek Bhardwaj[7,8], Steven J. Burgess[9], Iuliia Ferling[10], Małgorzata Anna Gazda[11,12], Luke H. Hoeppner[13,14], Vinodh Ilangovan[15], Hung Lo[16,17], Mischa Olson[18], Salem Yousef Mohamed[19], Sarvenaz Sarabipour[20], Aalok Varma[21], Kaivalya Walavalkar[21], Erin M. Wissink[22], Tracey L. Weissgerber[23]*

1 Mildred Scheel Early Career Center, Medical Faculty, Technische Universität Dresden, Dresden, Germany, 2 Department of Electronics, Information and Bioengineering, Politecnico di Milano, Italy, 3 Department of Brain and Behavioral Sciences, University of Pavia, Pavia, Italy, 4 Orthogonal Research and Education Laboratory, Champaign, IL, United States of America, 5 Evolutionary Genomics Unit, Okinawa Institute of Science and Technology, Okinawa, Japan, 6 Department of Plant Physiology, Faculty of Biology, Technische Universität Dresden, Dresden, Germany, 7 Max Plank Institute of Immunology and Epigenetics, Freiburg, Germany, 8 Hubrecht Institute, Utrecht, the Netherlands, 9 Carl R Woese Institute for Genomic Biology, University of Illinois at Urbana-Champaign, Urbana, IL, United States of America, 10 Junior Research Group Evolution of Microbial Interactions, Leibniz Institute for Natural Product Research and Infection Biology—Hans Knöll Institute (HKI), Jena, Germany, 11 CIBIO/InBIO, Centro de Investigação em Biodiversidade e Recursos Genéticos, Campus Agrário de Vairão, Universidade do Porto, Vairão, Portugal, 12 Departamento de Biologia, Faculdade de Ciências, Universidade do Porto, Porto, Portugal, 13 The Hormel Institute, University of Minnesota, Austin, MN, United States of America, 14 The Masonic Cancer Center, University of Minnesota, Minneapolis, MN, United States of America, 15 Aarhus University, Aarhus, Denmark, 16 Neuroscience Research Center, Charité—Universitätsmedizin Berlin, Corporate member of Freie Universität Berlin, Humboldt—Universität zu Berlin, Berlin Institute of Health, Berlin, Germany, 17 Einstein Center for Neurosciences Berlin, Berlin, Germany, 18 Section of Plant Biology, School of Integrative Plant Science, Cornell University, Ithaca, NY, United States of America, 19 Gastroenterology and Hepatology Unit, Internal Medicine Department, Faculty of Medicine, University of Zagazig, Zagazig, Egypt, 20 Institute for Computational Medicine and the Department of Biomedical Engineering, Johns Hopkins University, Baltimore, MD, United States of America, 21 National Centre for Biological Sciences (NCBS), Tata Institute of Fundamental Research (TIFR), Bangalore, Karnataka, India, 22 Department of Molecular Biology and Genetics, Cornell University, Ithaca, NY, United States of America, 23 Berlin Institute of Health at Charité–Universitätsmedizin Berlin, QUEST Center, Berlin, Germany

‡ These authors share first authorship on this work.
* tracey.weissgerber@charite.de

**Data Availability Statement:** The authors confirm that all data underlying the findings are fully available without restriction. The abstraction protocol, data, code and slides for teaching are

## Abstract

Scientists routinely use images to display data. Readers often examine figures first; therefore, it is important that figures are accessible to a broad audience. Many resources discuss fraudulent image manipulation and technical specifications for image acquisition; however, data on the legibility and interpretability of images are scarce. We systematically examined these factors in non-blot images published in the top 15 journals in 3 fields; plant sciences, cell biology, and physiology ($n = 580$ papers). Common problems included missing scale bars, misplaced or poorly marked insets, images or labels that were not accessible to color-blind readers, and insufficient explanations of colors, labels, annotations, or the species and tissue or object depicted in the image. Papers that met all good practice criteria examined for all image-based figures were uncommon (physiology 16%, cell biology 12%, plant sciences 2%). We present detailed descriptions and visual examples to help scientists avoid

available on an OSF repository (https://doi.org/10.17605/OSF.IO/B5296).

**Funding:** TLW was funded by American Heart Association grant 16GRNT30950002 (https://www.heart.org/en/professional/institute/grants) and a Robert W. Fulk Career Development Award (Mayo Clinic Division of Nephrology & Hypertension; https://www.mayoclinic.org/departments-centers/nephrology-hypertension/sections/overview/ovc-20464571). LHH was supported by The Hormel Foundation and National Institutes of Health grant CA187035 (https://www.nih.gov). The funders had no role in study design, data collection and analysis, decision to publish, or preparation of the manuscript.

**Competing interests:** The authors have declared that no competing interests exist.

**Abbreviations:** GFP, green fluorescent protein; LUT, lookup table; OSF, Open Science Framework; RRID, research resource identifier.

common pitfalls when publishing images. Our recommendations address image magnification, scale information, insets, annotation, and color and may encourage discussion about quality standards for bioimage publishing.

## Introduction

Images are often used to share scientific data, providing the visual evidence needed to turn concepts and hypotheses into observable findings. An analysis of 8 million images from more than 650,000 papers deposited in PubMed Central revealed that 22.7% of figures were "photographs," a category that included microscope images, diagnostic images, radiology images, and fluorescence images [1]. Cell biology was one of the most visually intensive fields, with publications containing an average of approximately 0.8 photographs per page [1]. Plant sciences papers included approximately 0.5 photographs per page [1].

While there are many resources on fraudulent image manipulation and technical requirements for image acquisition and publishing [2–4], data examining the quality of reporting and ease of interpretation for image-based figures are scarce. Recent evidence suggests that important methodological details about image acquisition are often missing [5]. Researchers generally receive little or no training in designing figures; yet many scientists and editors report that figures and tables are one of the first elements that they examine when reading a paper [6,7]. When scientists and journals share papers on social media, posts often include figures to attract interest. The PubMed search engine caters to scientists' desire to see the data by presenting thumbnail images of all figures in the paper just below the abstract [8]. Readers can click on each image to examine the figure, without ever accessing the paper or seeing the introduction or methods. EMBO's Source Data tool (RRID:SCR_015018) allows scientists and publishers to share or explore figures, as well as the underlying data, in a findable and machine readable fashion [9].

Image-based figures in publications are generally intended for a wide audience. This may include scientists in the same or related fields, editors, patients, educators, and grants officers. General recommendations emphasize that authors should design figures for their audience rather than themselves and that figures should be self-explanatory [7]. Despite this, figures in papers outside one's immediate area of expertise are often difficult to interpret, marking a missed opportunity to make the research accessible to a wide audience. Stringent quality standards would also make image data more reproducible. A recent study of fMRI image data, for example, revealed that incomplete documentation and presentation of brain images led to nonreproducible results [10,11].

Here, we examined the quality of reporting and accessibility of image-based figures among papers published in top journals in plant sciences, cell biology, and physiology. Factors assessed include the use of scale bars, explanations of symbols and labels, clear and accurate inset markings, and transparent reporting of the object or species and tissue shown in the figure. We also examined whether images and labels were accessible to readers with the most common form of color blindness [12]. Based on our results, we provide targeted recommendations about how scientists can create informative image-based figures that are accessible to a broad audience. These recommendations may also be used to establish quality standards for images deposited in emerging image data repositories.

## Results

### Using a science of science approach to investigate current practices

This study was conducted as part of a participant-guided learn-by-doing course, in which eLife Community Ambassadors from around the world worked together to design, complete,

and publish a meta-research study [13]. Participants in the 2018 Ambassadors program designed the study, developed screening and abstraction protocols, and screened papers to identify eligible articles (HJ, BA, SJB, VB, LHH, VI, SS, EMW). Participants in the 2019 Ambassadors program refined the data abstraction protocol, completed data abstraction and analysis, and prepared the figures and manuscript (AA, SA, TLA, IF, MAG, HL, SYM, MO, AV, KW, HJ, TLW).

To investigate current practices in image publishing, we selected 3 diverse fields of biology to increase generalizability. For each field, we examined papers published in April 2018 in the top 15 journals, which publish original research (S1–S3 Tables). All full-length original research articles that contained at least one photograph, microscope image, electron microscope image, or clinical image (MRI, ultrasound, X-ray, etc.) were included in the analysis (S1 Fig). Blots and computer-generated images were excluded, as some of the criteria assessed do not apply to these types of images. Two independent reviewers assessed each paper, according to the detailed data abstraction protocol (see methods and information deposited on the Open Science Framework (OSF) (RRID:SCR_017419) at https://doi.org/10.17605/OSF.IO/B5296) [14]. The repository also includes data, code, and figures.

## Image analysis

First, we confirmed that images are common in the 3 biology subfields analyzed. More than half of the original research articles in the sample contained images (plant science: 68%, cell biology: 72%, physiology: 55%). Among the 580 papers that included images, microscope images were very common in all 3 fields (61% to 88%, Fig 1A). Photographs were very common in plant sciences (86%), but less widespread in cell biology (38%) and physiology (17%). Electron microscope images were less common in all 3 fields (11% to 19%). Clinical images, such as X-rays, MRI or ultrasound, and other types of images were rare (2% to 9%).

Scale information is essential to interpret biological images. Approximately half of papers in physiology (49%) and cell biology (55%) and 28% of plant science papers provided scale bars with dimensions (in the figure or legend) for all images in the paper (Fig 1B, S4 Table). Approximately one-third of papers in each field contained incomplete scale information, such as reporting magnification or presenting scale information for a subset of images. Twenty-four percent of physiology papers, 10% of cell biology papers, and 29% of plant sciences papers contained no scale information on any image.

Some publications use insets to show the same image at 2 different scales (cell biology papers: 40%, physiology: 17%, plant sciences: 12%). In this case, the authors should indicate the position of the high-magnification inset in the low-magnification image. The majority of papers in all 3 fields clearly and accurately marked the location of all insets (53% to 70%; Fig 1C, left panel); however, one-fifth of papers appeared to have marked the location of at least one inset incorrectly (17% to 22%). Clearly visible inset markings were missing for some or all insets in 13% to 28% of papers (Fig 1C, left panel). Approximately half of papers (43% to 53%; Fig 1C, right panel) provided legend explanations or markings on the figure to clearly show that an inset was used, whereas this information was missing for some or all insets in the remaining papers.

Many images contain information in color. We sought to determine whether color images were accessible to readers with deuteranopia, the most common form of color blindness, by using the color blindness simulator Color Oracle (https://colororacle.org/, RRID: SCR_018400). We evaluated only images in which the authors selected the image colors (e.g., fluorescence microscopy). Papers without any colorblind accessible figures were uncommon (3% to 6%); however, 45% of cell biology papers and 21% to 24% of physiology and plant science papers contained some images that were inaccessible to readers with deuteranopia (Fig 2A). Seventeen

**A** Percentage of papers with image type

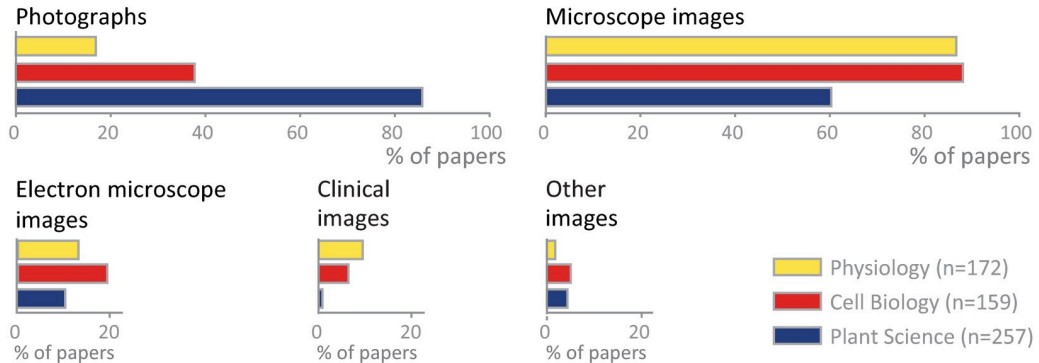

**B** Reporting of scale information

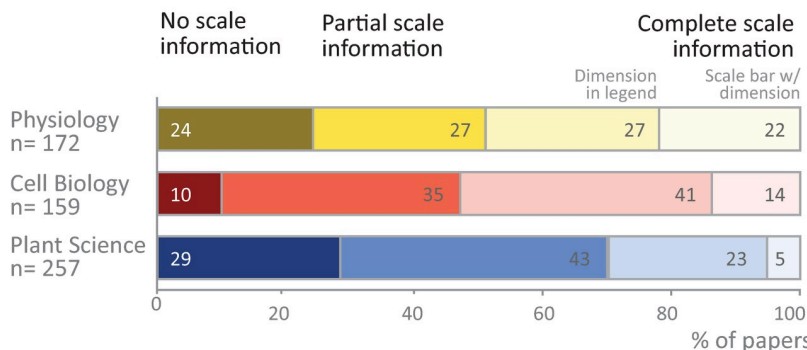

**C** Reporting of image insets

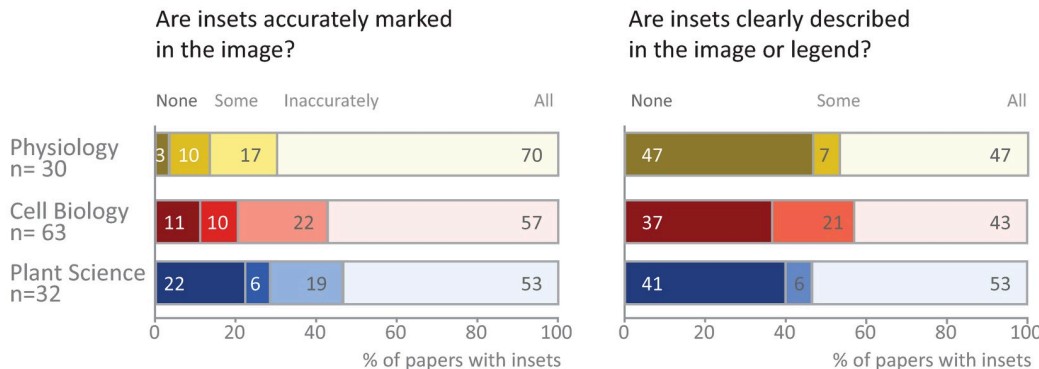

**Fig 1. Image types and reporting of scale information and insets. (A)** Microscope images and photographs were common, whereas other types of images were used less frequently. **(B)** Complete scale information was missing in more than half of the papers examined. Partial scale information indicates that scale information was presented in some figures, but not others, or that the authors reported magnification rather than including scale bars on the image. **(C)** Problems with labeling and describing insets are common. Totals may not be exactly 100% due to rounding.

**A** Colorblind accessibility

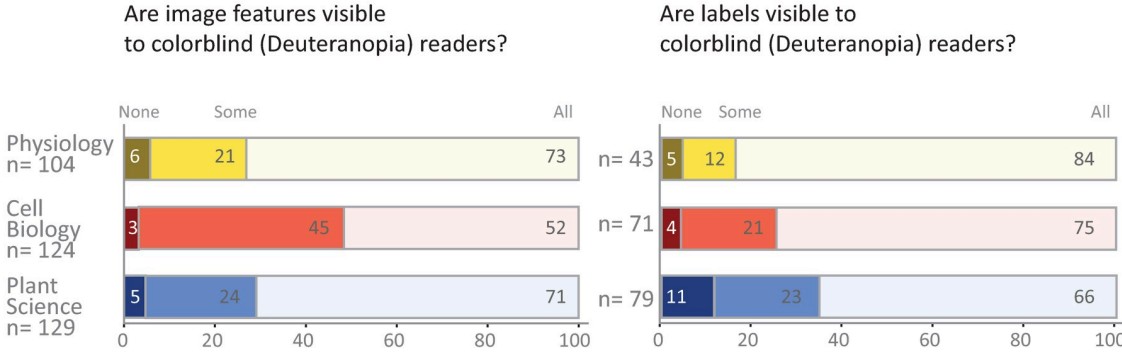

**B** Explanation in figure legends

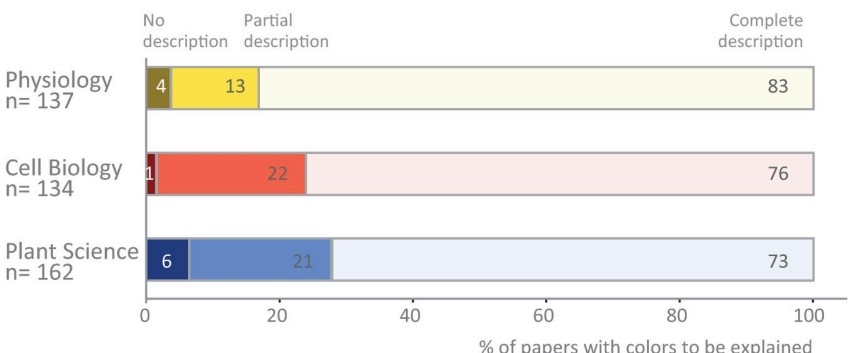

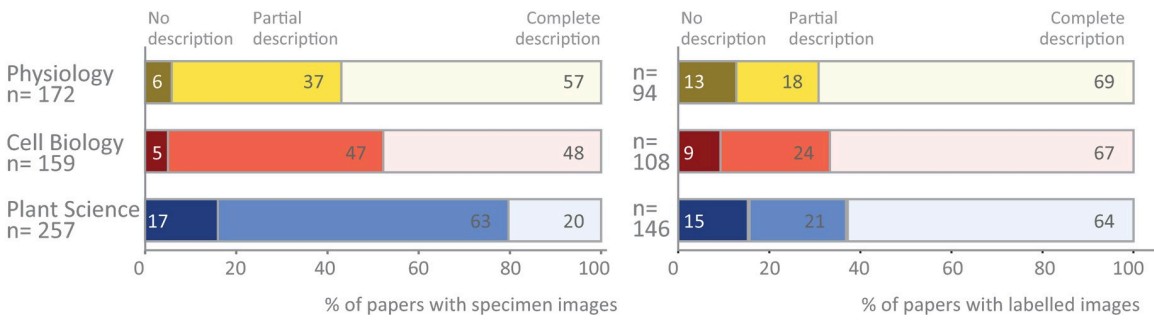

**Fig 2. Use of color and annotations in image-based figures. (A)** While many authors are using colors and labels that are visible to colorblind readers, the data show that improvement is needed. **(B)** Most papers explain colors in image-based figures; however, explanations are less common for the species and tissue or object shown, and labels and annotations. Totals may not be exactly 100% due to rounding.

percent to 34% of papers contained color annotations that were not visible to someone with deuteranopia.

Figure legends and, less often, titles typically provide essential information needed to interpret an image. This text provides information on the specimen and details of the image, while also explaining labels and annotations used to highlight structures or colors. Fifty-seven percent of physiology papers, 48% of cell biology papers, and 20% of plant papers described the species and tissue or object shown completely. Five percent to 17% of papers did not provide any such information (Fig 2B). Approximately half of the papers (47% to 58%; Fig 1C, right panel) also failed or partially failed to adequately explain that insets were used. Annotations of structures were better explained. Two-thirds of papers across all 3 fields clearly stated the meaning of all image labels, while 18% to 24% of papers provided partial explanations. Most papers (73% to 83%) completely explained the image colors by stating what substance each color represented or naming the dyes or staining technique used.

Finally, we examined the number of papers that used optimal image presentation practices for all criteria assessed in the study. Twenty-eight (16%) physiology papers, 19 (12%) cell biology papers, and 6 (2%) plant sciences papers met all criteria for all image-based figures in the paper. In plant sciences and physiology, the most common problems were with scale bars, insets, and specifying in the legend the species and tissue or object shown. In cell biology, the most common problems were with insets, colorblind accessibility, and specifying in the legend the species and tissue or object shown.

## Designing image-based figures: How can we improve?

Our results obtained by examining 580 papers from 3 fields provide us with unique insights into the quality of reporting and the accessibility of image-based figures. Our quantitative description of standard practices in image publication highlights opportunities to improve transparency and accessibility to readers from different backgrounds. We have therefore outlined specific actions that scientists can take when creating images, designing multipanel figures, annotating figures, and preparing figure legends.

Throughout the paper, we provide visual examples to illustrate each stage of the figure preparation process. Other elements are often omitted to focus readers' attention on the step illustrated in the figure. For example, a figure that highlights best practices for displaying scale bars may not include annotations designed to explain key features of the image. When preparing image-based figures in scientific publications, readers should address all relevant steps in each figure. All steps described below (image cropping and insets, adding scale bars and annotation, choosing color channel appearances, figure panel layout) can be implemented with standard image processing software such as FIJI [15] (RRID:SCR_002285) and ImageJ2 [16] (RRID: SCR_003070), which are open source, free programs for bioimage analysis. A quick guide on how to do basic image processing for publications with FIJI is available in a recent cheat sheet publication [17], and a discussion forum and wiki are available for FIJI and ImageJ (https:// imagej.net/).

**1. Choose a scale or magnification that fits your research question.** Scientists should select an image scale or magnification that allows readers to clearly see features needed to answer the research question. Fig 3A [18] shows *Drosophila melanogaster* at 3 different microscopic scales. The first focuses on the ovary tissue and might be used to illustrate the appearance of the tissue or show stages of development. The second focuses on a group of cells. In this example, the "egg chamber" cells show different nucleic acid distributions. The third example focuses on subcellular details in one cell, for example, to show finer detail of RNA granules or organelle shape.

**A** Magnification/zoom must match message

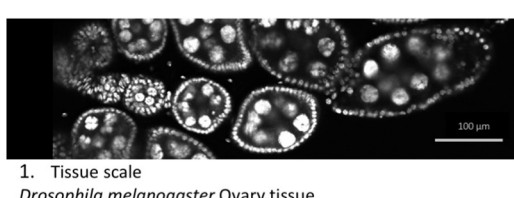

1.  Tissue scale
*Drosophila melanogaster* Ovary tissue

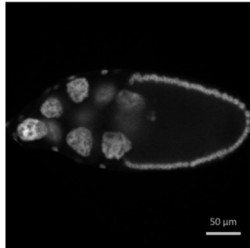
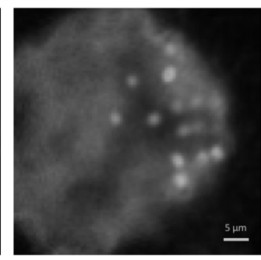

2. Cellular scale
*Drosophila melanogaster*
Egg chamber with oocyte

3. Subcellular scale
*Drosophila melanogaster*,
RNA granules in epithelial cell

**B** Insets allow readers to see more than one scale

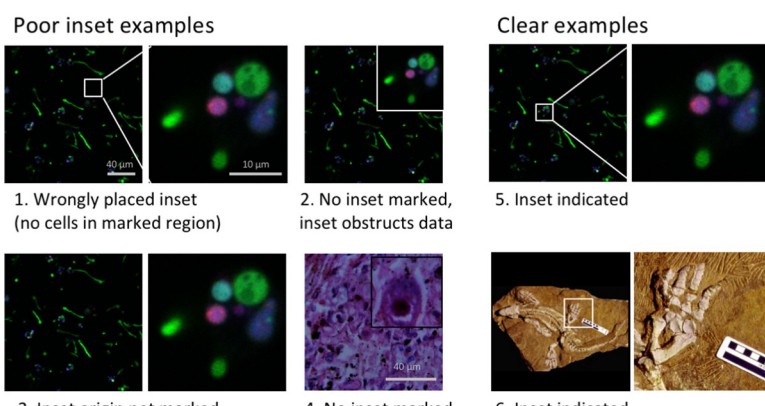

Poor inset examples

1. Wrongly placed inset
(no cells in marked region)

2. No inset marked,
inset obstructs data

3. Inset origin not marked

4. No inset marked,
inset obstructs data

Clear examples

5. Inset indicated

6. Inset indicated

**Fig 3. Selecting magnification and using insets. (A)** Magnification and display detail of images should permit readers to see features related to the main message that the image is intended to convey. This may be the organism, tissue, cell, or a subcellular level. Microscope images [18] show *D. melanogaster* ovary (A1), ovarian egg chamber cells (A2), and a detail in egg chamber cell nuclei (A3). **(B)** Insets or zoomed-in areas are useful when 2 different scales are needed to allow readers to see essential features. It is critical to indicate the origin of the inset in the full-scale image. Poor and clear examples are shown. Example images were created based on problems observed by reviewers. Images show B1, B2, B3, B5: *Protostelium aurantium* amoeba fed on germlings of *Aspergillus fumigatus* D141-GFP (green) fungal hyphae, dead fungal material stained with propidium iodide (red), and acidic compartments of amoeba marked with LysoTracker Blue DND-22 dye (blue); B4: Lendrum-stained human lung tissue (Haraszti, Public Health Image Library); B6: fossilized *Orobates pabsti* [19].

When both low and high magnifications are necessary for one image, insets are used to show a small portion of the image at higher magnification (Fig 3B, [19]). The inset location must be accurately marked in the low-magnification image. We observed that the inset position in the low-magnification image was missing, unclear, or incorrectly placed in approximately one-third of papers. Inset positions should be clearly marked by lines or regions of interest in a high-contrast color, usually black or white. Insets may also be explained in the

figure legend. Care must be taken when preparing figures outside vector graphics suits, as insert positions may move during file saving or export.

**2. Include a clearly labeled scale bar.** Scale information allows audiences to quickly understand the size of features shown in images. This is especially important for microscopic images where we have no intuitive understanding of scale. Scale information for photographs should be considered when capturing images as rulers are often placed into the frame. Our analysis revealed that 10% to 29% of papers screened failed to provide any scale information and that another third only provided incomplete scale information (Fig 1B). Scientists should consider the following points when displaying scale bars:

- **Every image type needs a scale bar:** Authors usually add scale bars to microscope images but often leave them out in photos and clinical images, possibly because these depict familiar objects such a human or plant. Missing scale bars, however, adversely affect reproducibility. A size difference of 20% in between a published study and the reader's lab animals, for example, could impact study results by leading to an important difference in phenotype. Providing scale bars allows scientists to detect such discrepancies and may affect their interpretation of published work. Scale bars may not be a standard feature of image acquisition and processing software for clinical images. Authors may need to contact device manufacturers to determine the image size and add height and width labels.

- **Scale bars and labels should be clearly visible:** Short scale bars, thin scale bars, and scale bars in colors that are similar to the image color can easily be overlooked (Fig 4). In multicolor images, it can be difficult to find a color that makes the scale bar stand out. Authors can solve this problem by placing the scale bar outside the image or onto a box with a more suitable background color.

Fig 4. **Using scale bars to annotate image size.** Scale bars provide essential information about the size of objects, which orients readers and helps them to bridge the gap between the image and reality. Scales may be indicated by a known size indicator such as a human next to a tree, a coin next to a rock, or a tape measure next to a smaller structure. In microscope images, a bar of known length is included. Example images were created based on problems observed by reviewers. Poor scale bar examples (1 to 6), clear scale bar examples (7 to 12). Images 1, 4, 7: Microscope images of *D. melanogaster* nurse cell nuclei [18]; 2: Microscope image of *Dictyostelium discoideum* expressing Vps32-GFP (Vps32-green fluorescent protein shows broad signal in cells) and stained with dextran (spotted signal) after infection with conidia of *Aspergillus fumigatus*; 3, 5, 8, 10: Electron microscope image of mouse pancreatic beta-islet cells (Andreas Müller); 6, 11: Microscope image of Lendrum-stained human lung tissue (Haraszti, Public Health Image Library); 9: Photo of *Arabidopsis thaliana*; 12: Photograph of fossilized *Orobates pabsti* [19].

- **Annotate scale bar dimensions on the image:** Stating the dimensions along with the scale bar allows readers to interpret the image more quickly. Despite this, dimensions were typically stated in the legend instead (Fig 1B), possibly a legacy of printing processes that discouraged text in images. Dimensions should be in high resolution and large enough to be legible. In our set, we came across small and/or low-resolution annotations that were illegible in electronic versions of the paper, even after zooming in. Scale bars that are visible on larger figures produced by authors may be difficult to read when the size of the figure is reduced to fit onto a journal page. Authors should carefully check page proofs to ensure that scale bars and dimensions are clearly visible.

 **3. Use color wisely in images.** Colors in images are used to display the natural appearance of an object or to visualize features with dyes and stains. In the scientific context, adapting colors is possible and may enhance readers' understanding, while poor color schemes may distract or mislead. Images showing the natural appearance of a subject, specimen, or staining technique (e.g., images showing plant size and appearance, or histopathology images of fat tissue from mice on different diets) are generally presented in color (Fig 5). Images showing electron microscope images are captured in black and white ("grayscale") by default and may be kept in grayscale to leverage the good contrast resulting from a full luminescence spectrum.

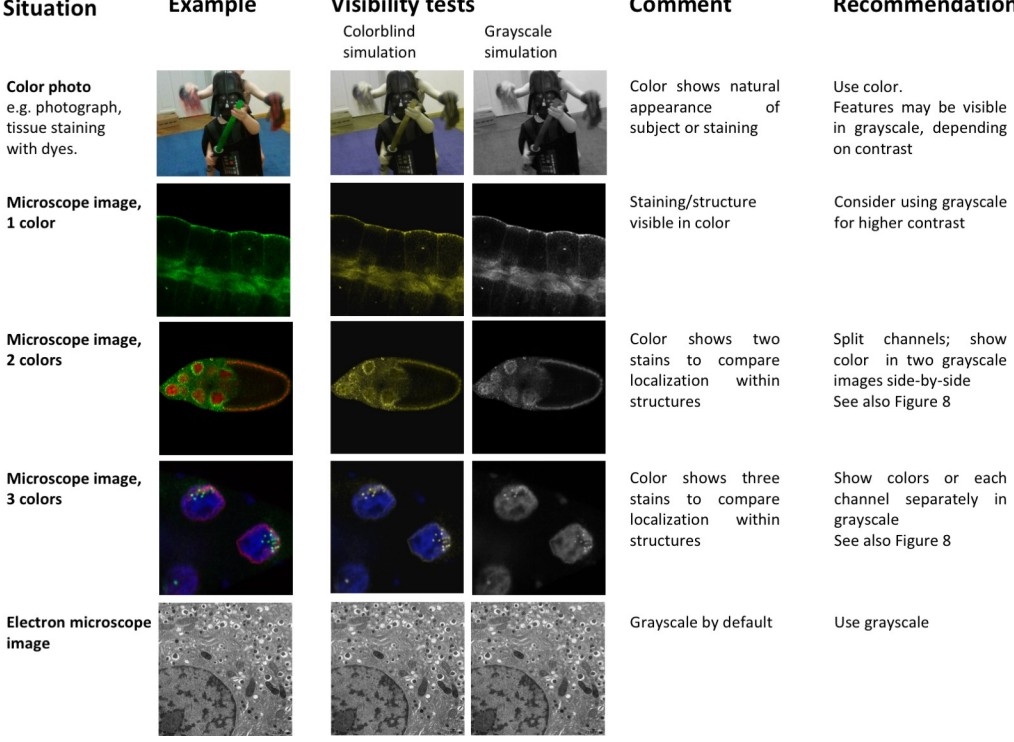

**Fig 5. Image types and their accessibility in colorblind render and grayscale mode.** Shown are examples of the types of images that one might find in manuscripts in the biological or biomedical sciences: photograph, fluorescent microscope images with 1 to 3 color hues/LUT, electron microscope images. The relative visibility is assessed in a colorblind rendering for deuteranopia, and in grayscale. Grayscale images offer the most contrast (1-color microscope image) but cannot show several structures in parallel (multicolor images, color photographs). Color combinations that are not colorblind accessible were used in rows 3 and 4 to illustrate the importance of colorblind simulation tests. Scale bars are not included in this figure, as they could not be added in a nondistracting way that would not detract from the overall message of the figure. Images show: Row 1: Darth Vader being attacked, Row 2: *D. melanogaster* salivary glands [18], Row 3: *D. melanogaster* egg chambers [18], Row 4: *D. melanogaster* nurse cell nuclei [18], and Row 5: mouse pancreatic beta-islet cells. LUT, lookup table.

**Color images**

**Grayscale test for visibility**

**Fig 6. Visibility of colors/hues differs and depends on the background color.** The best contrast is achieved with grayscale images or dark hues on a light background (first row). Dark color hues, such as red and blue, on a dark background (last row), are least visible. Visibility can be tested with mock grayscale. Images show actin filaments in *Dictyostelium discoideum* (LifeAct-GFP). All images have the same scale. GFP, green fluorescent protein.

In some instances, scientists can choose whether to show grayscale or color images. Assigning colors may be optional, even though it is the default setting in imaging programs. When showing only one color channel, scientists may consider presenting this channel in grayscale to optimally display fine details. This may include variations in staining intensity or fine structures. When opting for color, authors should use grayscale visibility tests (Fig 6) to determine whether visibility is compromised. This can occur when dark colors, such as magenta, red, or blue, are shown on a black background.

**4. Choose a colorblind accessible color palette.** Fluorescent images with merged color channels visualize the colocalization of different markers. While many readers find these images to be visually appealing and informative, these images are often inaccessible to colorblind coauthors, reviewers, editors, and readers. Deuteranopia, the most common form of colorblindness, affects up to 8% of men and 0.5% of women of northern European ancestry [12]. A study of articles published in top peripheral vascular disease journals revealed that 85% of papers with color maps and 58% of papers with heat maps used color palettes that were not colorblind safe [20]. We show that approximately half of cell biology papers, and one-third of physiology papers and plant science papers, contained images that were inaccessible to readers with deuteranopia. Scientists should consider the following points to ensure that images are accessible to colorblind readers.

- **Select colorblind safe colors:** Researchers should use colorblind safe color palettes for fluorescence and other images where color may be adjusted. Fig 7 illustrates how 4 different color combinations would look to viewers with different types of color blindness. Green and red are indistinguishable to readers with deuteranopia, whereas green and blue are indistinguishable to readers with tritanopia, a rare form of color blindness. Cyan and magenta are the best options, as these 2 colors look different to viewers with normal color vision, deuteranopia, or tritanopia. Green and magenta are also shown, as scientists often prefer to show colors close to the excitation value of the fluorescent dyes, which are often green and red.

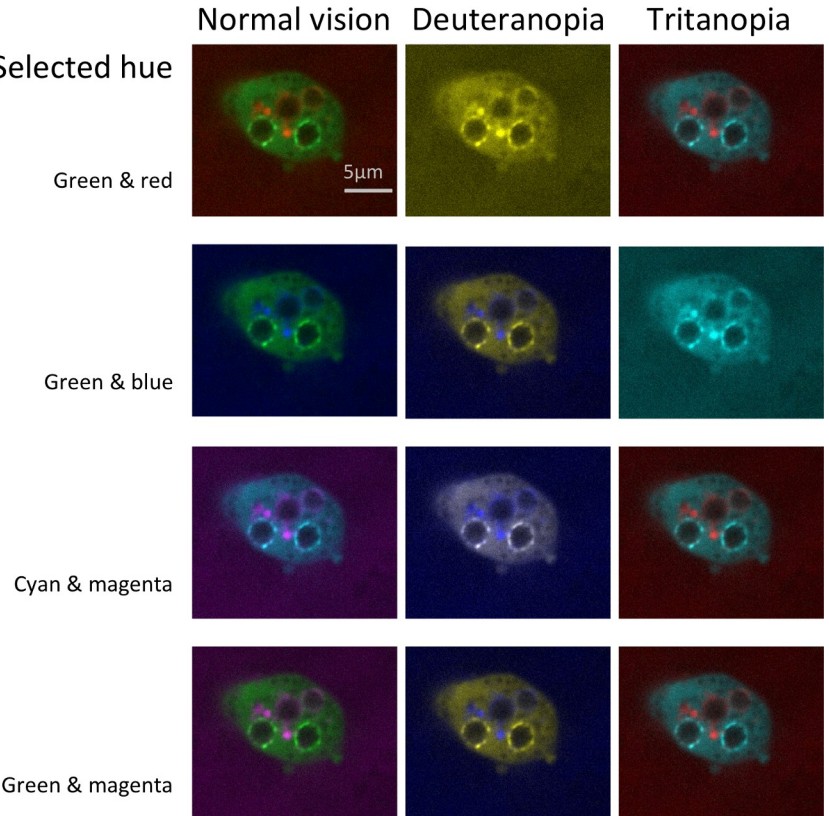

**Fig 7. Color combinations as seen with normal vision and 2 types of colorblindness.** The figure illustrates how 4 possible color combinations for multichannel microscope images would appear to someone with normal color vision, the most common form of colorblindness (deuteranopia), and a rare form of color blindness (tritanopia). Some combinations that are accessible to someone with deuteranopia are not accessible to readers with tritanopia, for example, green/blue combinations. Microscope images show *Dictyostelium discoideum* expressing Vps32-GFP (Vps32-green fluorescent protein shows broad signal in cells) and stained with dextran (spotted signal) after infection with conidia of *Aspergillus fumigatus*. All images have the same scale. GFP, green fluorescent protein.

- **Display separate channels in addition to the merged image:** Selecting a colorblind safe color palette becomes increasingly difficult as more colors are added. When the image includes 3 or more colors, authors are encouraged to show separate images for each channel, followed by the merged image (Fig 8). Individual channels may be shown in grayscale to make it easier for readers to perceive variations in staining intensity.

- **Use simulation tools to confirm that essential features are visible to colorblind viewers:** Free tools, such as Color Oracle (RRID:SCR_018400), quickly simulate different forms of color blindness by adjusting the colors on the computer screen to simulate what a colorblind person would see. Scientists using FIJI (RRID:SCR002285) can select the "Simulate color-blindness" option in the "Color" menu under "Images."

 **5. Design the figure.** Figures often contain more than one panel. Careful planning is needed to convey a clear message, while ensuring that all panels fit together and follow a logical order. A planning table (Fig 9A) helps scientists to determine what information is needed to answer the research question. The table outlines the objectives, types of visualizations required, and experimental groups that should appear in each panel. A planning table template is

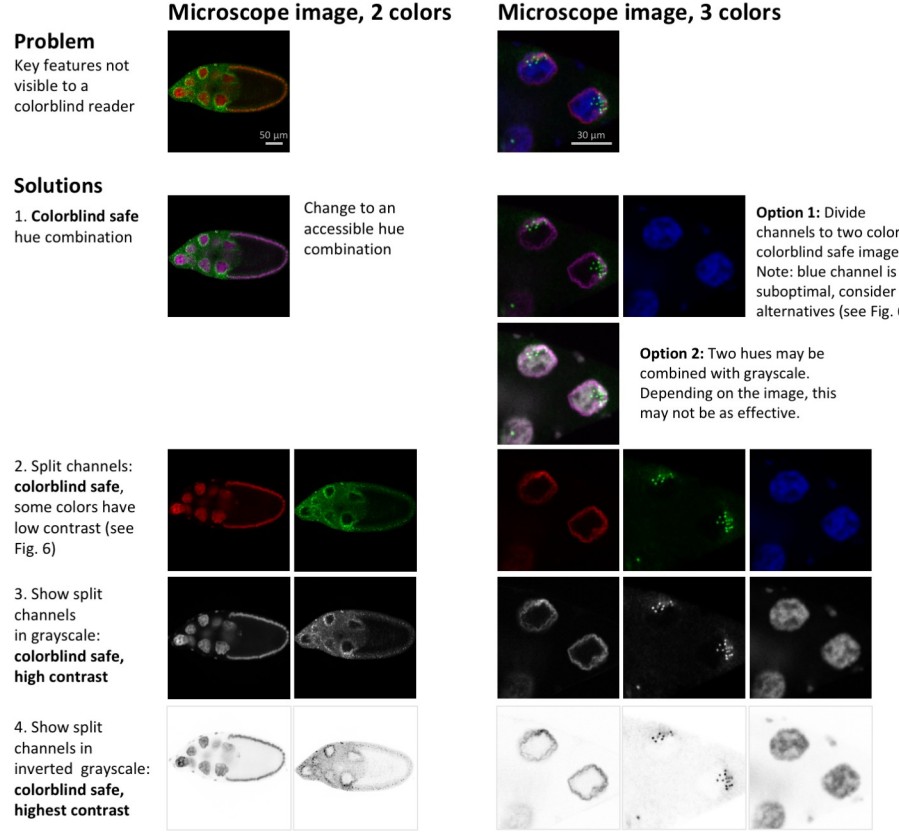

**Fig 8. Strategies for making 2- or 3-channel microscope images colorblind safe.** Images in the first row are not colorblind safe. Readers with the most common form of colorblindness would not be able to identify key features. Possible accessible solutions are shown: changing colors/LUTs to colorblind-friendly combinations, showing each channel in a separate image, showing colors in grayscale and inverting grayscale images to maximize contrast. Solutions 3 and 4 (show each channel in grayscale, or in inverted grayscale) are more informative than solutions 1 and 2. Regions of overlap are sometimes difficult to see in merged images without split channels. When splitting channels, scientists often use colors that have low contrast, as explained in Fig 6 (e.g., red or blue on black). Microscope images show *D. melanogaster* egg chambers (2 colors) and nurse cell nuclei (3 colors) [18]. All images of egg chambers and nurse cells respectively have the same scale. LUT, lookup table.

available on OSF [14]. After completing the planning table, scientists should sketch out the position of panels and the position of images, graphs, and titles within each panel (Fig 9B). Audiences read a page either from top to bottom and/or from left to right. Selecting one reading direction and arranging panels in rows or columns helps with figure planning. Using enough white space to separate rows or columns will visually guide the reader through the figure. The authors can then assemble the figure based on the draft sketch.

**6. Annotate the figure.** Annotations with text, symbols, or lines allow readers from many different backgrounds to rapidly see essential features, interpret images, and gain insight. Unfortunately, scientists often design figures for themselves, rather than their audience [7]. Examples of annotations are shown in Fig 10. Table 1 describes important factors to consider for each annotation type.

When adding annotations to an image, scientists should consider the following steps.

- **Choose the right amount of labeling.** Fig 11 shows 3 levels of annotation. The barely annotated image (Fig 11A) is only accessible to scientists already familiar with the object and technique, whereas the heavily annotated version (Fig 11C) contains numerous annotations that

**A** Organize and plan figures with a "**Figure planning table**"
Example for a study of mouse placenta genetics and test of a treatment.

| Panel | Panel objective | Visualizations | Experimental groups | Notes |
|---|---|---|---|---|
| A | Illustrate differences in pup phenotype | Photograph, chart | 1. Control group + placebo 2. Animal model + placebo 3. Control group + treatment 4. Animal model + treatment | Photo with scale (ruler for pups) Box plot: fetal weight |
| B | Illustrate differences in placenta phenotype | Photograph, chart | See above | Photo with scale (ruler) Box plot: placental weight |
| C | Illustrate histological differences in placenta e.g. staining for two biomarkers | Micrograph | See above | One image per group; separate rows for each biomarker |

**B** Organize panels into **"Figure layout sketch"**, exemplary for Figure planning table in A

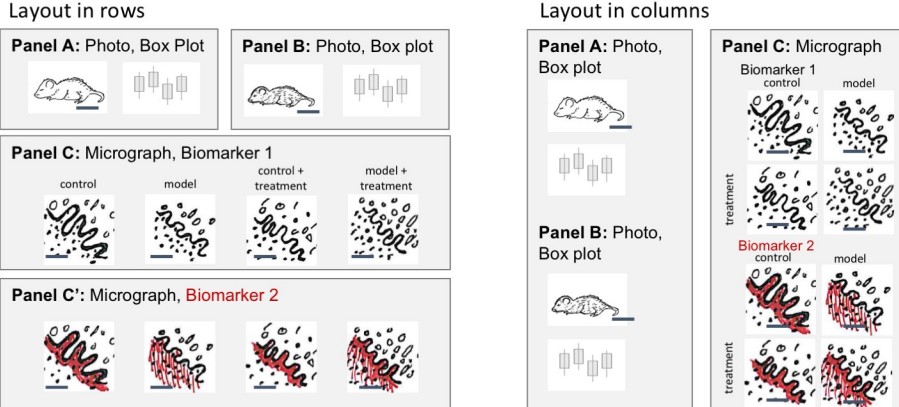

**Fig 9. Planning multipanel figures.** Planning tables and layout sketches are useful tools to efficiently design figures that address the research question. (**A**) Planning tables allow scientists to select and organize elements needed to answer the research question addressed by the figure. (**B**) Layout sketches allow scientists to design a logical layout for all panels listed in the planning table and ensure that there is adequate space for all images and graphs.

obstruct the image and a legend that is time consuming to interpret. Fig 11B is more read-able; annotations of a few key features are shown, and the explanations appear right below the image for easy interpretation. Explanations of labels are often placed in the figure legend. Alternating between examining the figure and legend is time consuming, especially when the legend and figure are on different pages. Fig 11D shows one option for situations where extensive annotations are required to explain a complex image. An annotated image is placed as a legend next to the original image. A semitransparent white layer mutes the image to allow annotations to stand out.

- **Use abbreviations cautiously:** Abbreviations are commonly used for image and figure annotation to save space but inevitably require more effort from the reader. Abbreviations are often ambiguous, especially across fields. Authors should run a web search for the abbreviation [21]. If the intended meaning is not a top result, authors should refrain from using the abbreviation or clearly define the abbreviation on the figure itself, even if it is already defined elsewhere in the manuscript. Note that in Fig 11, abbreviations have been written out below the image to reduce the number of legend entries.

- **Explain colors and stains:** Explanations of colors and stains were missing in around 20% of papers. Fig 12 illustrates several problematic practices observed in our dataset, as well as

## Annotation strategies

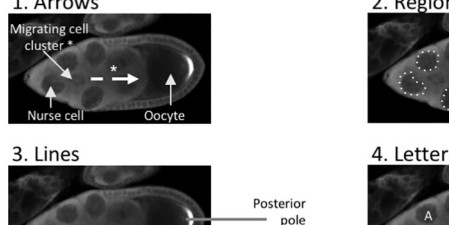

| | Use | Caution | Tips |
|---|---|---|---|
| 1. Arrows | Points to structure<br>May indicate direction of movement (e.g. *) | Do not concurrently use arrows for pointing to structure and indicating movement (example *)<br>Arrowheads alone often have no clear direction | Avoid crossing arrows<br>Align arrows |
| 2. Region of interest | Delineates entire structure | May obstruct image features (especially when fill color is used) | Careful when saving: dashed lines may be too thin |
| 3. Lines | Direct labeling of structure at line end | Label may be outside of image to not obstruct image features | Avoid crossing lines<br>Align lines |
| 4. Letter code, symbol | Labels many features clearly where lines and arrows would confuse | Legend is critical, requires large space.<br>Labels may obscure image features | Choose suitable font e.g. sans serif |

**Fig 10. Using arrows, regions of interest, lines, and letter codes to annotate structures in images.** Text descriptions alone are often insufficient to clearly point to a structure or region in an image. Arrows and arrowheads, lines, letters, and dashed enclosures can help if overlaid on the respective part of the image. Microscope images show *D. melanogaster* egg chambers [18], with the different labeling techniques in use. The table provides an overview of their applicability and common pitfalls. All images have the same scale.

**Table 1. Use annotations to make figures accessible to a broad audience.**

| Feature to be Explained | Annotation |
|---|---|
| Size | Scale bar with dimensions |
| Direction of movement | Arrow with tail |
| Draw attention to: | |
| • Points of interest | Symbol (arrowhead, star, etc.) |
| • Regions of interest: black and white image | Highlight in color if this does not obscure important features within the region OR<br>Outline with boxes or circles |
| • Regions of interest: Color image | Outline with boxes or circles |
| • Layers | Labeled brackets beside the image for layers that are visually identifiable across the entire image OR<br>A line on the image for wavy layers that may be difficult to identify |
| Define features within an image | Labels |

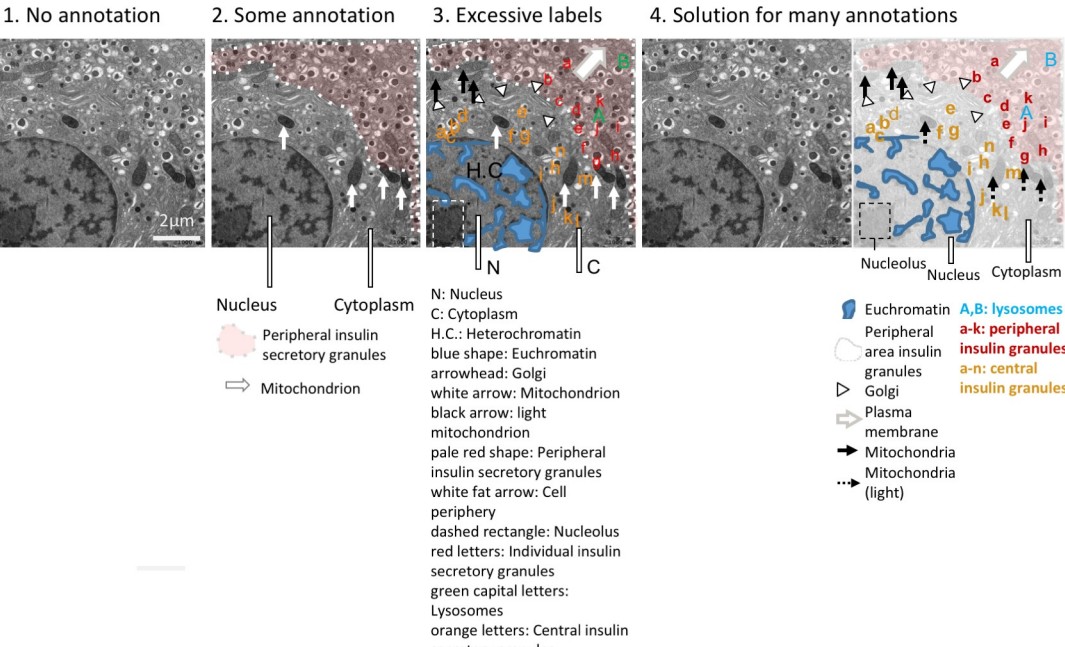

**Fig 11. Different levels of detail for image annotations.** Annotations help to orient the audience but may also obstruct parts of the image. Authors must find the right balance between too few and too many annotations. (1) Example with no annotations. Readers cannot determine what is shown. (2) Example with a few annotations to orient readers to key structures. (3) Example with many annotations, which obstruct parts of the image. The long legend below the figure is confusing. (4) Example shows a solution for situations where many annotations are needed to explain the image. An annotated version is placed next to an unannotated version of the image for comparison. The legend below the image helps readers to interpret the image, without having to refer to the figure legend. Note the different requirements for space. Electron microscope images show mouse pancreatic beta-islet cells.

solutions for clearly explaining what each color represents. This figure uses fluorescence images as an example; however, we also observed many histology images in which authors did not mention which stain was used. Authors should describe how stains affect the tissue shown or use annotations to show staining patterns of specific structures. This allows readers who are unfamiliar with the stain to interpret the image.

- **Ensure that annotations are accessible to colorblind readers:** Confirming that labels or annotations are visible to colorblind readers is important for both color and grayscale images (Fig 13). Up to one-third of papers in our dataset contained annotations or labels that would not have been visible to someone with deuteranopia. This occurred because the annotations blended in with the background (e.g., red arrows on green plants) or the authors use the same symbol in colors that are indistinguishable to someone with deuteranopia to mark different features. Fig 13 illustrates how to annotate a grayscale image so that it is accessible to color blind readers. Using text to describe colors is also problematic for colorblind readers. This problem can be alleviated by using colored symbols in the legend or by using distinctly shaped annotations such as open versus closed arrows, thin versus wide lines, or dashed versus solid lines. Color blindness simulators help in determining whether annotations are accessible to all readers.

  **7. Prepare figure legends.** Each figure and legend are meant to be self-explanatory and should allow readers to quickly assess a paper or understand complex studies that combine different methodologies or model systems. To date, there are no guidelines for figure legends for

**Poor color annotation**

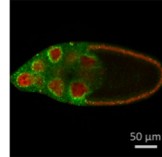
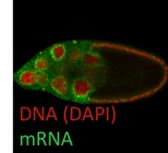
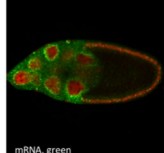
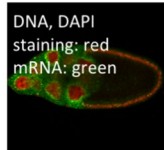

1. No color annotation

2. Color annotation not colorblind safe

3. Illegible and/or incomplete annotation

4. Annotation covers image content

**Clear color annotation**

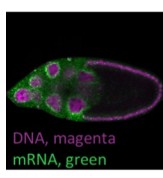
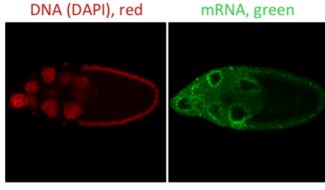
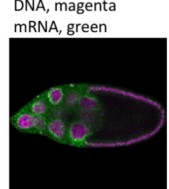

5. Colorblind safe annotation

6. Colorblind image and annotation (split color channels). Note: red has low contrast on black background, see Fig. 6.

7. Name colors when text must be in grayscale

**Fig 12. Explain color in images.** Cells and their structures are almost all transparent. Every dye, stain, and fluorescent label therefore should be clearly explained to the audience. Labels should be colorblind safe. Large labels that stand out against the background are easy to read. Authors can make figures easier to interpret by placing the color label close to the structure; color labels should only be placed in the figure legend when this is not possible. Example images were created based on problems observed by reviewers. Microscope images show *D. melanogaster* egg chambers stained with the DNA dye DAPI (4′,6-diamidino-2-phenylindole) and probe for a specific mRNA species [18]. All images have the same scale.

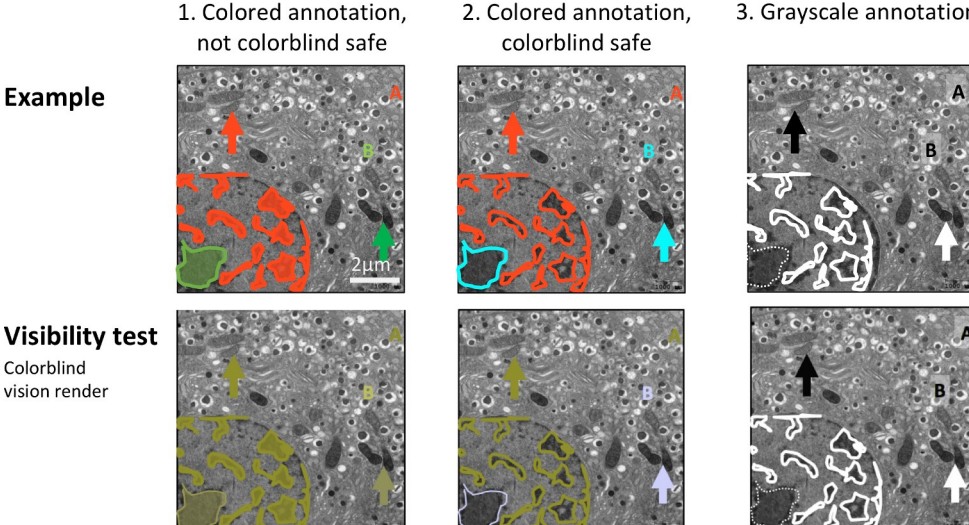

**Fig 13. Annotations should be colorblind safe.** (1) The annotations displayed in the first image are inaccessible to colorblind individuals, as shown with the visibility test below. This example was created based on problems observed by reviewers. (2, 3) Two colorblind safe alternative annotations, in color (2) and in grayscale (3). The bottom row shows a test rendering for deuteranopia colorblindness. Note that double-encoding of different hues and different shapes (e.g., different letters, arrow shapes, or dashed/nondashed lines) allows all audiences to interpret the annotations. Electron microscope images show mouse pancreatic beta-cell islet cells. All images have the same scale.

images, as the scope and length of legends varies across journals and disciplines. Some journals require legends to include details on object, size, methodology, or sample size, while other journals require a minimalist approach and mandate that information should not be repeated in subsequent figure legends.

Our data suggest that important information needed to interpret images was regularly missing from the figure or figure legend. This includes the species and tissue type, or object shown in the figure, clear explanations of all labels, annotations and colors, and markings or legend entries denoting insets. Presenting this information on the figure itself is more efficient for the reader; however, any details that are not marked in the figure should be explained in the legend.

While not reporting species and tissue information in every figure legend may be less of an issue for papers that examine a single species and tissue, this is a major problem when a study includes many species and tissues, which may be presented in different panels of the same figure. Additionally, the scientific community is increasingly developing automated data mining tools, such as the Source Data tool, to collect and synthesize information from figures and other parts of scientific papers. Unlike humans, these tools cannot piece together information scattered throughout the paper to determine what might be shown in a particular figure panel. Even for human readers, this process wastes time. Therefore, we recommend that authors present information in a clear and accessible manner, even if some information may be repeated for studies with simple designs.

## Discussion

A flood of images is published every day in scientific journals and the number is continuously increasing. Of these, around 4% likely contain intentionally or accidentally duplicated images [3]. Our data show that, in addition, most papers show images that are not fully interpretable due to issues with scale markings, annotation, and/or color. This affects scientists' ability to interpret, critique, and build upon the work of others. Images are also increasingly submitted to image archives to make image data widely accessible and permit future reanalyses. A substantial fraction of images that are neither human nor machine-readable lowers the potential impact of such archives. Based on our data examining common problems with published images, we provide a few simple recommendations, with examples illustrating good practices. We hope that these recommendations will help authors to make their published images legible and interpretable.

**Limitations:** While most results were consistent across the 3 subfields of biology, findings may not be generalizable to other fields. Our sample included the top 15 journals that publish original research for each field. Almost all journals were indexed in PubMed. Results may not be generalizable to journals that are unindexed, have low impact factors, or are not published in English. Data abstraction was performed manually due to the complexity of the assessments. Error rates were 5% for plant sciences, 4% for physiology, and 3% for cell biology. Our assessments focused on factors that affect readability of image-based figures in scientific publications. Future studies may include assessments of raw images and meta-data to examine factors that affect reproducibility, such as contrast settings, background filtering, and processing history.

### Actions journals can take to make image-based figures more transparent and easier to interpret

The role of journals in improving the quality of reporting and accessibility of image-based figures should not be overlooked. There are several actions that journals might consider.

- **Screen manuscripts for figures that are not colorblind safe:** Open source automated screening tools [22] may help journals to efficiently identify common color maps that are not colorblind safe.

- **Update journal policies:** We encourage journal editors to update policies regarding color-blind accessibility, scale bars, and other factors outlined in this manuscript. Importantly, policy changes should be accompanied by clear plans for implementation and enforcement. Meta-research suggests that changing journal policy, without enforcement or implementation plans, has limited effects on author behavior. Amending journal policies to require authors to report research resource identifiers (RRIDs), for example, increases the number of papers reporting RRIDs by 1% [23]. In a study of life sciences articles published in Nature journals, the percentage of animal studies reporting the Landis 4 criteria (blinding, randomization, sample size calculation, exclusions) increased from 0% to 16.4% after new guidelines were released [24]. In contrast, a randomized controlled trial of animal studies submitted to *PLOS ONE* demonstrated that randomizing authors to complete the ARRIVE checklist during submission did not improve reporting [25]. Some improvements in reporting of confidence intervals, sample size justification, and inclusion and exclusion criteria were noted after Psychological Science introduced new policies [26], although this may have been partially due to widespread changes in the field. A joint editorial series published in the Journal of Physiology and British Journal of Pharmacology did not improve the quality of data presentation or statistical reporting [27].

- **Reevaluate limits on the number of figures:** Limitations on the number of figures originally stemmed from printing costs calculations, which are becoming increasingly irrelevant as scientific publishing moves online. Unintended consequences of these policies include the advent of large, multipanel figures. These figures are often especially difficult to interpret because the legend appears on a different page, or the figure combines images addressing different research questions.

- **Reduce or eliminate page charges for color figures:** As journals move online, policies designed to offset the increased cost of color printing are no longer needed. The added costs may incentivize authors to use grayscale in cases where color would be beneficial.

- **Encourage authors to explain labels or annotations in the figure, rather than in the legend:** This is more efficient for readers.

- **Encourage authors to share image data in public repositories:** Open data benefits authors and the scientific community [28–30].

## How can the scientific community improve image-based figures?

The role of scientists in the community is multifaceted. As authors, scientists should familiarize themselves with guidelines and recommendations, such as ours provided above. As reviewers, scientists should ask authors to improve erroneous or uninformative image-based figures. As instructors, scientists should ensure that bioimaging and image data handling is taught during undergraduate or graduate courses, and support existing initiatives such as NEUBIAS (Network of EUropean BioImage AnalystS) [31] that aim to increase training opportunities in bioimage analysis.

Scientists are also innovators. As such, they should support emerging image data archives, which may expand to automatically source images from published figures. Repositories for other types of data are already widespread; however, the idea of image repositories has only

recently gained traction [32]. Existing image databases, which are mainly used for raw image data and meta-data, include the Allen Brain Atlas, the Image Data Resource [33], and the emerging BioImage Archives [32]. Springer Nature encourages authors to submit imaging data to the Image Data Resource [33]. While scientists have called for common quality standards for archived images and meta-data [32], such standards have not been defined, implemented, or taught. Examining standard practices for reporting images in scientific publications, as outlined here, is one strategy for establishing common quality standards.

In the future, it is possible that each image published electronically in a journal or submitted to an image data repository will follow good practice guidelines and will be accompanied by expanded "meta-data" or "alt-text/attribute" files. Alt-text is already published in html to provide context if an image cannot be accessed (e.g., by blind readers). Similarly, images in online articles and deposited in archives could contain essential information in a standardized format. The information could include the main objective of the figure, specimen information, ideally with RRID [34], specimen manipulation (dissection, staining, RRID for dyes and antibodies used), as well as the imaging method including essential items from meta-files of the microscope software, information about image processing and adjustments, information about scale, annotations, insets, and colors shown, and confirmation that the images are truly representative.

## Conclusions

Our meta-research study of standard practices for presenting images in 3 fields highlights current shortcomings in publications. Pubmed indexes approximately 800,000 new papers per year, or 2,200 papers per day (https://www.nlm.nih.gov/bsd/index_stats_comp.html). Twenty-three percent [1], or approximately 500 papers per day, contain images. Our survey data suggest that most of these papers will have deficiencies in image presentation, which may affect legibility and interpretability. These observations lead to targeted recommendations for improving the quality of published images. Our recommendations are available as a slide set via the OSF and can be used in teaching best practice to avoid misleading or uninformative image-based figures. Our analysis underscores the need for standardized image publishing guidelines. Adherence to such guidelines will allow the scientific community to unlock the full potential of image collections in the life sciences for current and future generations of researchers.

## Methods

### Systematic review

We examined original research articles that were published in April of 2018 in the top 15 journals that publish original research for each of 3 different categories (physiology, plant science, cell biology). Journals for each category were ranked according to 2016 impact factors listed for the specified categories in Journal Citation Reports. Journals that only publish review articles or that did not publish an April issue were excluded. We followed all relevant aspects of the PRISMA guidelines [35]. Items that only apply to meta-analyses or are not relevant to literature surveys were not followed. Ethical approval was not required.

### Search strategy

Articles were identified through a PubMed search, as all journals were PubMed indexed. Electronic search results were verified by comparison with the list of articles published in April issues on the journal website. The electronic search used the following terms:

Physiology: ("Journal of pineal research"[Journal] AND 3[Issue] AND 64[Volume]) OR ("Acta physiologica (Oxford, England)"[Journal] AND 222[Volume] AND 4[Issue]) OR ("The Journal of physiology"[Journal] AND 596[Volume] AND (7[Issue] OR 8[Issue])) OR (("American journal of physiology. Lung cellular and molecular physiology"[Journal] OR "American journal of physiology. Endocrinology and metabolism"[Journal] OR "American journal of physiology. Renal physiology"[Journal] OR "American journal of physiology. Cell physiology"[Journal] OR "American journal of physiology. Gastrointestinal and liver physiology"[Journal]) AND 314[Volume] AND 4[Issue]) OR ("American journal of physiology. Heart and circulatory physiology"[Journal] AND 314[Volume] AND 4[Issue]) OR ("The Journal of general physiology"[Journal] AND 150[Volume] AND 4[Issue]) OR ("Journal of cellular physiology"[Journal] AND 233[Volume] AND 4[Issue]) OR ("Journal of biological rhythms"[Journal] AND 33[Volume] AND 2[Issue]) OR ("Journal of applied physiology (Bethesda, Md.: 1985)"[Journal] AND 124[Volume] AND 4[Issue]) OR ("Frontiers in physiology"[Journal] AND ("2018/04/01"[Date—Publication]: "2018/04/30"[Date—Publication])) OR ("The international journal of behavioral nutrition and physical activity"[Journal] AND ("2018/04/01"[Date—Publication]: "2018/04/30"[Date—Publication])).

Plant science: ("Nature plants"[Journal] AND 4[Issue] AND 4[Volume]) OR ("Molecular plant"[Journal] AND 4[Issue] AND 11[Volume]) OR ("The Plant cell"[Journal] AND 4[Issue] AND 30[Volume]) OR ("Plant biotechnology journal"[Journal] AND 4[Issue] AND 16[Volume]) OR ("The New phytologist"[Journal] AND (1[Issue] OR 2[Issue]) AND 218[Volume]) OR ("Plant physiology"[Journal] AND 4[Issue] AND 176[Volume]) OR ("Plant, cell & environment"[Journal] AND 4[Issue] AND 41[Volume]) OR ("The Plant journal: for cell and molecular biology"[Journal] AND (1[Issue] OR 2[Issue]) AND 94[Volume]) OR ("Journal of experimental botany"[Journal] AND (8[Issue] OR 9[Issue] OR 10[Issue]) AND 69[Volume]) OR ("Plant & cell physiology"[Journal] AND 4[Issue] AND 59[Volume]) OR ("Molecular plant pathology"[Journal] AND 4[Issue] AND 19[Volume]) OR ("Environmental and experimental botany"[Journal] AND 148[Volume]) OR ("Molecular plant-microbe interactions: MPMI"[Journal] AND 4[Issue] AND 31[Volume]) OR ("Frontiers in plant science"[Journal] AND ("2018/04/01"[Date—Publication]: "2018/04/30"[Date—Publication])) OR ("The Journal of ecology" ("2018/04/01"[Date—Publication]: "2018/04/30"[Date—Publication])).

Cell biology: ("Cell"[Journal] AND (2[Issue] OR 3[Issue]) AND 173[Volume]) OR ("Nature medicine"[Journal] AND 24[Volume] AND 4[Issue]) OR ("Cancer cell"[Journal] AND 33[Volume] AND 4[Issue]) OR ("Cell stem cell"[Journal] AND 22[Volume] AND 4[Issue]) OR ("Nature cell biology"[Journal] AND 20[Volume] AND 4[Issue]) OR ("Cell metabolism"[Journal] AND 27[Volume] AND 4[Issue]) OR ("Science translational medicine"[Journal] AND 10[Volume] AND (435[Issue] OR 436[Issue] OR 437[Issue] OR 438[Issue])) OR ("Cell research"[Journal] AND 28[Volume] AND 4[Issue]) OR ("Molecular cell"[Journal] AND 70[Volume] AND (1[Issue] OR 2[Issue])) OR("Nature structural & molecular biology"[Journal] AND 25[Volume] AND 4[Issue]) OR ("The EMBO journal"[Journal] AND 37[Volume] AND (7[Issue] OR 8[Issue])) OR ("Genes & development"[Journal] AND 32[Volume] AND 7–8[Issue]) OR ("Developmental cell"[Journal] AND 45[Volume] AND (1[Issue] OR 2[Issue])) OR ("Current biology: CB"[Journal] AND 28[Volume] AND (7[Issue] OR 8[Issue])) OR ("Plant cell"[Journal] AND 30[Volume] AND 4[Issue]).

## Screening

Screening for each article was performed by 2 independent reviewers (Physiology: TLW, SS, EMW, VI, KW, MO; Plant science: TLW, SJB; Cell biology: EW, SS) using Rayyan software (RRID:SCR_017584), and disagreements were resolved by consensus. A list of articles was

uploaded into Rayyan. Reviewers independently examined each article and marked whether the article was included or excluded, along with the reason for exclusion. Both reviewers screened all articles published in each journal between April 1 and April 30, 2018, to identify full length, original research articles (S1–S3 Tables, S1 Fig) published in the print issue of the journal. Articles for online journals that do not publish print issues were included if the publication date was between April 1 and April 30, 2018. Articles were excluded if they were not original research articles, or if an accepted version of the paper was posted as an "in press" or "early release" publication; however, the final version did not appear in the print version of the April issue. Articles were included if they contained at least one eligible image, such as a photograph, an image created using a microscope or electron microscope, or an image created using a clinical imaging technology such as ultrasound or MRI. Blot images were excluded, as many of the criteria in our abstraction protocol cannot easily be applied to blots. Computer generated images, graphs, and data figures were also excluded. Papers that did not contain any eligible images were excluded.

## Abstraction

All abstractors completed a training set of 25 articles before abstracting data. Data abstraction for each article was performed by 2 independent reviewers (Physiology: AA, AV; Plant science: MO, TLA, SA, KW, MAG, IF; Cell biology: IF, AA, AV, KW, MAG). When disagreements could not be resolved by consensus between the 2 reviewers, ratings were assigned after a group review of the paper. Eligible manuscripts were reviewed in detail to evaluate the following questions according to a predefined protocol (available at: https://doi.org/10.17605/OSF.IO/B5296) [14]. Supplemental files were not examined, as supplemental images may not be held to the same peer review standards as those in the manuscript.

The following items were abstracted:

1. Types of images included in the paper (photograph, microscope image, electron microscope image, image created using a clinical imaging technique such as ultrasound or MRI, other types of images)

2. Did the paper contain appropriately labeled scale bars for all images?

3. Were all insets clearly and accurately marked?

4. Were all insets clearly explained in the legend?

5. Is the species and tissue, object, or cell line name clearly specified in the figure or legend for all images in the paper?

6. Are any annotations, arrows, or labels clearly explained for all images in the paper?

7. Among images where authors can control the colors shown (e.g., fluorescence microscopy), are key features of the images visible to someone with the most common form of color-blindness (deuteranopia)?

8. If the paper contains colored labels, are these labels visible to someone with the most common form of color blindness (deuteranopia)?

9. Are colors in images explained either on the image or within the legend?

Questions 7 and 8 were assessed by using Color Oracle [36] (RRID:SCR_018400) to simulate the effects of deuteranopia.

## Verification

Ten percent of articles in each field were randomly selected for verification abstraction, to ensure that abstractors in different fields were following similar procedures. Data were abstracted by a single abstractor (TLW). The question on species and tissue was excluded from verification abstraction for articles in cell biology and plant sciences, as the verification abstractor lacked the field-specific expertise needed to assess this question. Results from the verification abstractor were compared with consensus results from the 2 independent abstractors for each paper, and discrepancies were resolved through discussion. Error rates were calculated as the percentage of responses for which the abstractors' response was incorrect. Error rates were 5% for plant sciences, 4% for physiology, and 3% for cell biology.

## Data processing and creation of figures

Data are presented as n (%). Summary statistics were calculated using Python (RRID: SCR_008394, version 3.6.9, libraries NumPy 1.18.5 and Matplotlib 3.2.2). Charts were prepared with a Python-based Jupyter Notebook (Jupyter-client, RRID:SCR_018413 [37], Python version 3.6.9, RRID:SCR_008394, libraries NumPy 1.18.5 [38], and Matplotlib 3.2.2 [39]) and assembled into figures with vector graphic software. Example images were previously published or generously donated by the manuscript authors as indicated in the figure legends. Image acquisition was described in references (*D. melanogaster* images [18], mouse pancreatic beta islet cells: A. Müller personal communication, and *Orobates pabsti* [19]). Images were cropped, labeled, and color-adjusted with FIJI [15] (RRID:SCR_002285) and assembled with vector-graphic software. Colorblind and grayscale rendering of images was done using Color Oracle [36] (RRID:SCR_018400). All poor and clear images presented here are "mock examples" prepared based on practices observed during data abstraction.

## Supporting information

**S1 Fig. Flow chart of study screening and selection process.** This flow chart illustrates the number of included and excluded journals or articles, along with reasons for exclusion, at each stage of the study.
(JPG)

**S1 Table. Number of articles examined by journal in physiology.** Values are n, or n (% of all articles). Screening was performed to exclude articles that were not full-length original research articles (e.g., reviews, editorials, perspectives, commentaries, letters to the editor, short communications, etc.), were not published in April 2018, or did not include eligible images. AJP, American Journal of Physiology.
(DOCX)

**S2 Table. Number of articles examined by journal in plant science.** Values are n, or n (% of all articles). Screening was performed to exclude articles that were not full-length original research articles (e.g., reviews, editorials, perspectives, commentaries, letters to the editor, short communications, etc.), were not published in April 2018, or did not include eligible images. *This journal was also included on the cell biology list (Table S3). **No articles from the Journal of Ecology were screened as the journal did not publish an April 2018 issue.
(DOCX)

**S3 Table. Number of articles examined by journal in cell biology.** Values are n, or n (% of all articles). Screening was performed to exclude articles that were not full-length original research articles (e.g., reviews, editorials, perspectives, commentaries, letters to the editor,

short communications, etc.), were not published in April 2018, or did not include eligible images. *This journal was also included on the plant science list (Table S2).
(DOCX)

**S4 Table. Scale information in papers.** Values are percent of papers.
(DOCX)

## Acknowledgments

We thank the eLife Community Ambassadors program for facilitating this work, and Andreas Müller and John A. Nyakatura for generously sharing example images. Falk Hillmann and Thierry Soldati provided the amoeba strains used for imaging. Some of the early career researchers who participated in this research would like to thank their principal investigators and mentors for supporting their efforts to improve science.

## Author Contributions

**Conceptualization:** Helena Jambor, Tracey L. Weissgerber.

**Data curation:** Alberto Antonietti, Tracey L. Weissgerber.

**Formal analysis:** Alberto Antonietti.

**Investigation:** Helena Jambor, Alberto Antonietti, Bradly Alicea, Tracy L. Audisio, Susann Auer, Vivek Bhardwaj, Steven J. Burgess, Iuliia Ferling, Małgorzata Anna Gazda, Luke H. Hoeppner, Vinodh Ilangovan, Hung Lo, Mischa Olson, Salem Yousef Mohamed, Sarvenaz Sarabipour, Aalok Varma, Kaivalya Walavalkar, Erin M. Wissink, Tracey L. Weissgerber.

**Methodology:** Helena Jambor, Alberto Antonietti, Bradly Alicea, Tracy L. Audisio, Susann Auer, Vivek Bhardwaj, Steven J. Burgess, Iuliia Ferling, Małgorzata Anna Gazda, Luke H. Hoeppner, Vinodh Ilangovan, Hung Lo, Mischa Olson, Salem Yousef Mohamed, Sarvenaz Sarabipour, Aalok Varma, Kaivalya Walavalkar, Erin M. Wissink, Tracey L. Weissgerber.

**Project administration:** Tracey L. Weissgerber.

**Resources:** Helena Jambor, Susann Auer, Iuliia Ferling, Tracey L. Weissgerber.

**Supervision:** Tracey L. Weissgerber.

**Validation:** Tracey L. Weissgerber.

**Visualization:** Helena Jambor, Alberto Antonietti, Iuliia Ferling, Tracey L. Weissgerber.

**Writing – original draft:** Helena Jambor, Tracey L. Weissgerber.

**Writing – review & editing:** Helena Jambor, Alberto Antonietti, Bradly Alicea, Tracy L. Audisio, Susann Auer, Vivek Bhardwaj, Steven J. Burgess, Iuliia Ferling, Małgorzata Anna Gazda, Luke H. Hoeppner, Vinodh Ilangovan, Hung Lo, Mischa Olson, Salem Yousef Mohamed, Sarvenaz Sarabipour, Aalok Varma, Kaivalya Walavalkar, Erin M. Wissink, Tracey L. Weissgerber.

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
