## [Editor Report · Decision Letter 0]

28 Oct 2020

Dear Dr Weissgerber, 

Thank you for submitting your manuscript entitled "Creating Clear and Informative Image-based Figures for Scientific Publications" for consideration as a Meta-Research Article by PLOS Biology.

Your manuscript has now been evaluated by the PLOS Biology editorial staff as well as by an academic editor with relevant expertise and I am writing to let you know that we would like to send your submission out for external peer review.

Please re-submit your manuscript within two working days, i.e. by Oct 30 2020 11:59PM.

Once your full submission is complete, your paper will undergo a series of checks in preparation for peer review, after which it will be sent out for review. 

Given the disruptions resulting from the ongoing COVID-19 pandemic, please expect some delays in the editorial process. We apologise in advance for any inconvenience caused and will do our best to minimize impact as far as possible.

Kind regards,

Senior Editor

PLOS Biology

---

## [Decision Letter · Decision Letter 1]

9 Dec 2020

Dear Dr Weissgerber,

Thank you very much for submitting your manuscript "Creating Clear and Informative Image-based Figures for Scientific Publications" for consideration as a Meta-Research Article at PLOS Biology. Your manuscript has been evaluated by the PLOS Biology editors, an Academic Editor with relevant expertise, and by five independent reviewers. I must apologise for the excessive number of reviewers; we usually aim for three or four, but an administrative oversight led to us recruiting an extra one. I hope that you nevertheless find all the comments useful.

You'll see that the reviewers are broadly positive about your study, but each raises a number of concerns and makes suggestions for improvement. In light of the reviews (below), we are pleased to offer you the opportunity to address the from the reviewers in a revised version that we anticipate should not take you very long. We will then assess your revised manuscript and your response to the reviewers' comments and we may consult the reviewers again.

We expect to receive your revised manuscript within 1 month.

**IMPORTANT - SUBMITTING YOUR REVISION**

*Resubmission Checklist*

*Published Peer Review*

*PLOS Data Policy*

*Blot and Gel Data Policy*

Sincerely,

Roli Roberts

Senior Editor,

rroberts@plos.org,

PLOS Biology

REVIEWERS' COMMENTS:

Reviewer #1: 

[identifies herself as Elisabeth Bik]

In this paper, the authors screened hundreds of papers from three different scientific fields (physiology, cell biology, and plant sciences) and selected 580 papers that included photographic images. They analyzed the papers containing photographic images for the presence of scale bars, inset annotation, clear labeling, colorblindness-friendly color scheme, adequate description of the specimen etc. The majority of the papers failed one of these criteria. Examples of good and bad image labeling are given throughout the manuscript.

The paper is a welcome addition to the field of meta-science (science about science papers, and provides clear guidelines about what constitutes good labeling and color-use of photographic images in biomedical papers. The search strategy is clearly described and reproducible, and the paper was easy to read and understand. Also, kudos to the authors for including an image featuring Darth Vader. 

I have some minor comments.

General comments:

It would be nice if the Abstract should include the total number of papers (580) screened for this study - that number is somewhat hard to find. It is included in Figure S1 (flow chart) and the discussion but it would be good to include it in the abstract and the first paragraph of the Results (see below).

The term "Microphotograph" might benefit from a definition. It appears the authors mean a photo taken from a specimen under a microscope (e.g. of cells or tissues), but I am not sure. Is a "Photograph" then defined as a photo of something visible to the eye such as a plant or a petridish? One could call all the image types mentioned in Figure 1A "photographs", so maybe consider using the term "macrophotograph" for a photo that is not a microphotograph. 

Are the examples shown in Figure 4-6 from the papers that were screened for this paper? Or were they taken from public sources (as indicated for some photos) and then manipulated digitally to either remove or add a scale bar (see fig 4)? It would be nice to clearly define that in the Methods (or maybe I missed that).

Specific comments

Page 1, Affiliations of the authors: Typo: "Uterecht"

Introduction. At the end of the Introduction, and the end of "Using a science of science approach...." on Page 4, there are several references to specific figures. I would personally not expect these in the Results, but rather in the Introduction, so maybe consider removing part of that last paragraph of "Using a science...." to the beginning of the Results?

Results. Page 4. It would be more clear to start the Results section by mentioning how many papers (580) were screened. 

Results. Page 4. "More than half of the papers in the sample contained images (plant science: 68%, cell biology: 72%, physiology: 55%)." - These numbers do not seem to match the data provided in Supplemental Tables 1-3. Maybe I am misunderstanding something, but Supplemental Tables 1-3 mention 39.9, 51.2, and 38.9% of papers, which are much lower numbers. 

Physiology: 431 screened - 172 included (39.9%)

Plant science: 502 screened - 257 included (51.2%)

Cell Biology: 409 screened - 159 included (38.9%)

On page 6, "Approximately half of the papers (47-58%) also failed or partially failed to adequately explain insets. " appears to refer to Figure 1C, right panel, but the figure number/panel is not mentioned. Maybe add that?

Page 11, under 3 "Use Color wisely in images", "Images showing ielectron micrographs" should perhaps read "Images showing electron microphotographs"

Page 13, Maybe write "Deuteranopia, the most common form of colorblindness..." to remind the reader of what the term means (used a lot in the following paragraph)

Discussion. Page 22: "intentionally or accidentally manipulated images" - should be "intentionally or accidentally duplicated images"

Page 22: What is meant by "Error rates" here? The numbers listed here do not appear to match anything else in the paper. Maybe a reference or reminder needs to be included here.

Discussion. Page 22: "Actions journals can take to make image-based figures more transparent and easier to interpret". An important item not listed here, but that I personally think is very important, is to add particular requirements about e.g. the use of colorblind-safe colors and inclusion of scale bars in the journal's guidelines for figure preparation/guidelines for authors. Many of these requirements could be listed to the guidelines that many journals already have online. It is much easier to have these requirements up front instead of trying to fix them during the manuscript reviewing stage.

Page 23. "of which 500 are estimated to contain images" - do the authors mean photographic images? What is this number based on?

Figure 1B and Figure 1C layout could be more similar to each other

Figure 1C - right hand panel not described in Results, and not clear how it differs from what is shown in the left panel

In Figure 4, Square = 1cm, should this be 1cm2?

Figure 4 refers to 1-3 and 4-6 but there are no numbers in the figure itself. 

Figure 4 typo: "Micropcope"

Figure 12: In top right, I did not think the color annotation was that clear ; I liked the solution used in the top left, although that is not color blind safe - could something similar be used in the top right? The line to the mRNA appears to land in an area that has both colors, which was not very clear. Maybe moving it a bit to the left so that it would land in a clear green area would help.

Methods. Page 25, under "Screening" what is meant by "using Rayyan software"? I was not familiar with that tool. 

Supplemental materials. The Plant Cell articles were included twice in Tables S2 and S3, which was potentially confusing, since now the totals of Tables S1-S3 cannot be summed. I would recommend leaving them out of the Cell Biology table (S3), with a little note under the table, so that there are no duplicate values across the tables.

Table S1-S3: maybe include percentages in the top row, e.g., n=409 n=159 (38.9%)

Page 29, under Table S2, should be "This journal was also included on the cell biology list (Table S3)." instead of "(Table S2)".

Reviewer #2:

In general, I find this paper to be excellent and to be potentially a very valuable resource to the community. I appreciate the large amount of work their initial quantitative findings must have required, and the thoroughness of the recommendations they have put together.

My largest critique (the only one I feel would be NECESSARY to address before publication is that in general), the authors prescribe certain things readers should do when authoring their own papers, but are inconsistent in whether or not they tell readers how to do that (or point them to an educational resource). This is not universal- they do, for example, point the reader to resources for simulating colorblindness in the text around Figures 7 and 8, but not how to do the inversions or greyscale testing in Figure 6, how to generate labels ala Figures 10 and 11, etc. Obviously it would be outside the scope of this paper to teach readers to do every task in every POSSIBLE software it could be done in, but the authors could select one or two commonly used tools (such as FIJI, Photoshop/Illustrator, etc, though for maximum utility my vote would be for something free to use) and provide guidance in those. This could be done along the way, and/or as part of a section at the beginning describing what are some commonly used tools for figure creation (and pointing to resources for each to learn to do common tasks). In that vein it would also be nice for the authors to more fully credit the tools that were used to make their own figures (they describe which python libraries are used in the creation of their bar graphs, but don't cite the relevant publications for those libraries or for the jupyter project itself (which according to the OSF project is how those figures were created), nor do they describe which software tool(s) they used to create the rest of the figures (They mention the QuickFigures tool at one point, though it's not clear that is what's used in this work or not).

An additional few smaller critiques- 

1) The degree to which the authors obey their own rules for best practices vary; many of the images in the paper lack scale bars, for example, or have illegible bars (figure 6). I understand in most cases that is not the point being illustrated in that particular figure, and would not see it as a blocker for publication, but it would be nice to see them used more consistently, especially in the "good" images.

2) The text in the table in Figure 10 is VERY small, it might be better to move it below rather than beside the figure so it can more easily be enlarged. The text in other figures (such as 9 and 11 is also borderline tiny)

3) I personally find the broken-up-bar-graph in figure 1B a bit hard to read, especially as the bars for "Some scale bar dimensions" and "All/some magnification in legend" are overlapping; breaking it into multiple bar plots ala 1A lacks the "nice" effect of seeing how things add to 100 but might be more clear.

Reviewer #3:

The manuscript starts with quantification of image usage in publications and is followed by quantification of correct/incorrect image reporting (usage of scale information, insets etc.). The analysis of the published papers served the authors to discover problems and to come-up with suggestions that are presented in the following - core part of the manuscript. Here the authors give clear suggestions to relevant steps of image representation and figure preparation. Each step is visualized comparing wrong and right/improved approaches, such that the readers can compare the differences immediately by themselves. The manuscript ends with a final discussion that includes action points suggested to journal and the scientific community. The manuscript is very clearly written and gives the reader clear recommendations on how to improve image display. 

Novelty and significance

While the single steps addressed (scale bar, color scheme, annotations) are not novel, the way of presenting it with the comparison in figures and the focus on the "colorblind safe" images is. The discussion in context of modern publishing (online) and the connection to online image repositories is timely.

The manuscript gives the reader a very clear "workflow" of what to do in different cases (e.g. 2 color image vs. 3 color image, or EM image vs. color photo) in order to avoid pitfalls. With this I expect it to be of great use, especially (but not only) for early career scientists.

Points of criticism:

I would have wished for a discussion around the flexibility of the rules and a potential of "miscounting" in the quantification of fig 1. E.g. also in this manuscript the scale bar is missing in most figures and would have been counted accordingly as "Partial scale information" in figure 1. (The reason why the scale bar is missing is written in the text of the manuscript.)

Also, I would have wished for a discussion whether/whether not it is important to include details in the figure legend, especially about tissue specification. Under section 7 (prepare figure legends) it is written that some journals require details, while others not - which clearly shows different opinions about this topic. Figure 2B "Are species/tissue/object clearly described in the legend?" shows to me rather different opinions on this topic rather than clear errors in image representation.

Minor comments:

- Fig 1: Include to the supplementary examples of images classified as e.g. "insets inaccurately marked, some marked " etc. if this is possible following copyright of already published figures. 

- Fig 3A, subcellular scale image is saturated

- Fig 3B. Solution (cell image): inset marking is not fully transparent

- Fig 4: Ruler as scale bar - Square: 1cm; square not visible in this magnification

- Fig. 5: "Darth Vader being attached" - kids playing Star Wars?

- Section 5. Design the figure: "either from top to bottom and/or from right to left" should presumably read as "left to right"

- Fig 6 scale bar not visible in the print as it is for now

- Fig 8 Split the color channel: blue described as "least visible" in Fig. 6, but used anyway

- Same in Fig. 12 (red), described as "least visible" in Fig. 6, but used anyway

Reviewer #4:

[identifies herself as Perrine Paul-Gilloteaux]

 This paper proposes a systematic review of figures in literature in biology-related fields, following some of the PRISMA guidelines, to assess the quality of these published figures. The criteria assessed are the accessibility of figures for color-blindness scientist, the presence of some minimal information as defined by the authors in the legend, the clarity of annotations or insets as assessed by the authors, the presence and clarity of the scale bar. The minimal information (in addition to the scale bar) that should be reported in the legend, as defined by the authors, are defined as the species (or cell lines) observed and the explanation of colors shown. Statistics on the binary fulfilment of these criteria are reported on the selected sample of publications.

The main message reported is that a majority of figures manually inspected by the authors did not fulfil all these criteria. 

In addition the authors provides some examples of DO and DON'T for these points and provide guidelines to design good quality figures, according to these criteria.

While the study is certainly a considerable amount of work, and may point out that editors and reviewers did not do their job (PLOS Biology was not assessed) (reporting scale bar is at least known and required to be present and all figures by editors), I am questioning the choice of the criteria assessed. In particular, authors stated that these criteria serve the reproducibility, I do not understand how badly presented insets may reduce the reproducibility, as stated by the authors. It may unserve the readability, or send a bad message of the rigour of the study, but even this would need to be supported as statements, since in the study the figures which were not filling these criteria did not need them to be understood by the reader. More important guidelines, such as the one asked by journal publishing guidelines (contrast settings, background filtering, process history) would be more important as they can lead to wrong and false messages. The choice of these particular criteria should have been defended against some data or example about how they prevent reproducibility. 

Then, showing with the permission of editors/authors, some example of badly assessed figures would have been useful: in particular I am doubtful about the unvisible annotation due to the blending with background color and how it can escape, the example shown of DON'T would serve better the message if taken from real published papers. Real example from real papers of figures assessed as not filling some of the criteria would serve better the message of the paper. Or even more ambitious, adding some reporting on the subjective loss of information and understanding in these papers by the authors of this meta analysis?

For example, even if it is indeed not deserving the main message of the paper, scale bar is not reported in most of the figures of this paper itself (it would have been expected at least for the example of different scale of images Figure 3 ) and in the same time species is reported for all figures when it brings no element to the main message, which is not biologically-related.

Also in the reporting of the method, I could not get how was defined the error rate mentioned: discrepancy in the binary answer of reviewers on each criteria? Are the scripts to compute the statistics provided? I could not find it on the link provided by the authors.

In addition, one of the main conclusion is also that these recommendations could help in designing the minimal information required when depositing data, but actually the repositories mentioned (IDR, Cell Atlas) store the raw data, not the figures, so the criteria and factors assessed are not applicable. Could the authors comment on this point or clarify this?

In conclusion, while the topic is timely relevant in the time of the reproducibility crisis, the authors are sending some messages that should be in the hand of the editors while editing the final proof of papers, in particular with the limited amount and impact of the criteria assessed. The two parts of the paper: constatation of the state of figure published in April 2018 against the criteria defined by the authors, followed by related guidelines and recommendations, are coherent together but the angle taken is too narrow:, in particular when stating as a main mission the reproducibility of papers. It may be of relevance for teaching courses but I am not sure about its categorization as a research paper as it is. The meta analysis could be of further interest if the support of the message was stronger by proving how this failure in criteria deserves reproducibility and interpretation of the data, as I am not convinced the ones chosen are the more important.

Reviewer #5:

[identifies himself as Simon F. Nørrelykke]

 * Summary of the research and my overall impression

** 1. summarise what the ms claims to report

This manuscript details the results from a group of researchers across the globe who got together to document the state of image-based figures in scientific publications. The results obtained show that there is ample room for improvement and the authors proceed by giving figure-creation recommendations that, if followed by authors and journals, should greatly increase the quality of published figures.

Fraudulent image manipulation and how to acquire images is not the focus of this manuscript. Microscopy images, both transmitted, fluorescent, and electron, as well and photographs, are the focus; medical images (MRI, ultrasound, etc) were allowed but rare in the three fields studied. 

All papers published during April 2018 in 15 journals (45 journals in total) in the three fields of plant science, cell biology, and physiology were manually examined and scored along several dimensions according to a shared protocol, available online and discussed in the manuscript.

580 papers were examined by "eLife Community Ambassadors from around the world" working together.

Only 2--16% of these papers met all the criteria set for good practices.

Detailed recommendations are given for the preparation of figures with microscopy images. These include discussions of scale bars, insets, colors/colorblindness, label, annotations, legends etc.

Though figures are ideally be designed to reach a wide audience, incl. scientists in other fields, they are typically only interpretable by a very narrow one, if at all.

The advise given on selecting the relevant magnification, how and where to include scale bars, and usage of color, should all be common sense, but apparently is not (behold the results of the investigation reported in this manuscript.) They are thus valuable, even if not novel or thought-provoking, and should be mandatory reading for every student preparing their first manuscript - and perhaps for a majority of PIs, reviewers, and editors alike.

** 2. give overview of the strengths and weaknesses of the ms

- Well written manuscript that reads well (except, perhaps, for the results section)

- The results section is very dry. Six paragraphs lists a large number of percentages. This is data but almost not information. An actuarian may disagree. Figures contain slightly more data and in a more digestible format (graphical). 

- Data-acquisition: The number of journals assessed and the approach taken (two reviewers per paper and a clear protocol) is scientific and convincing

- The recommendations are clear and well illustrated

- Though most/all of the points are not new to anyone used to working with images (colorblindness, contrast, scale bars etc), it is useful to see them all collected and commented on in one place - also, every number of years it is useful to remind the community that these things are still (or increasingly? we don't know) an issue. 

- Being literal about PLOS criteria:

 + Originality :: this is, as far as I know, the first papers reporting solidly on image-based figure quality 

 + Importance to researchers in its field :: Important enough that it should be mandatory reading for any figure-creating scientist

 + Interest to scientists outside the field :: The findings and recommendations cover three fields and easily generalise to other fields

 + Rigorous methodology and substantial evidence for its conclusions :: Yes! Details given elsewhere in report.

** 3. recommended course of action

Publish after revision.

Highlight with editorial mention and Twitter activity.

This paper may do more for science than many a pure research manuscript.

* Specific areas that could be improved

** Major issues

- Major, somewhat, because pointing to conceptual issues

 + p. 6 "We evaluated only images in which the authors could have adjusted the image colors (i.e. fluorescence microscopy)"

 + Unless I misunderstand, it is perfectly possible to adjust the colors in any image, so this limitation to fluorescent microscopy images seems to not be justified by the argument given.

 + Example: In an RGB image, e.g. a photo of a flower, the user can set a different color for each of the three channels. This is easily done in, e.g. Imagej/Fiji using the channel tool

 * https://imagej.net/docs/guide/146-28.html#toc-Subsection-28.5

 * https://imagej.net/docs/guide/146-28.html#sub:Channels...[Z]

 + Fix: redo research or reformulate sentence to simply state which images you comment on. 

 + Or, did you perhaps mean "e.g." and not "i.e."?

- Major, but fixable, because pointing to conceptual issues

 + p. 12: "Digital microscope setups capture each channel's intensities in greyscale values."

 + Nope: Some do, some don't.

 + Fluorescent microscopes equipped with filter cubes and very light sensitive CCDs (CMOSs) tend to, as do confocals

 + Slides scanners (also microscopes) are usually equipped with RGB cameras.

 + Suggested fix: delete sentence after understanding why it is wrong

- Suggestion for how to lead by example and in the interest of reproducibility

 + Share the data in an interoperable manner (FAIR principles)

 + Share the Python notebooks used for statistical analysis

 + Share the scripts used to create figures (unless assembled by hand)

 + Do this in GitHub, Zenodo, or the journal website

** Minor issues

- p3: EMBO's Source Data tool (RRID:SCR_015018)

 + Is this supposed to be a link or reference?

- p6: "Color Oracle (https://colororacle.org/, RRID:SCR_018400)."

 + What is RRID? Not explained until p. 23.

- p. 5, Figure 1

 + Please give n in subpanel B, similar to A and C, or Fig 2 A, B, C.

 + Or state that numbers are the same as in A

- p. 11, Figure 4

 + This figure would be more powerful if the problems were 1-1 mirrored by solutions

 + Only two of the five problem images are solved

 + The ruler shown in the bottom right corner is too small to illustrate the point otherwise made: Zooming in, in the pdf, does not give clearly resolved 1cm squares, perhaps due to jpg effect.

 + Alternatively, rename from "problem" and "solution" to something not evoking expectations of solutions to the problems, e.g. by removing those two words.

- p. 12, Figure 5, top row

 + This is a very unlikely example of a scientific image

 + Resist temptation of including photos of family members ;-)

 + If you cannot find a natural, scientific, example, perhaps this is not an actual problem?

- p. 12, Figure 5, third and fourth row

 + Recommendations: the splitting should be in addition to, not instead of, adjusting for colorblindness in a merged image

 + Yes, you refer to Fig 8, but here is a natural place to mention it

- p. 13, Figure 6

 + This figure ought to be redundant, to the extent that the reader knows that higher contrast has higher contrast

 + If, however, the authors saw many examples of dark colors on dark background during their scans of papers, this could still seem a justified figure

- p. 14

 + "Free tools, such as Color Oracle (RRID:SCR_018400)"

 + Also available, for images, in the very popular open source software Fiji under "Image > Color > Simulate Color Blindness"

- p. 15, Figure 8

 + You show possible solutions but do not say what you recommend.

 + Please, do that and argue for the choice!

- p. 16

 + "QuickFigures (RRID:SCR019082)"

 + Does this software support reproducibility (creates scripts that can generate entire figure)?

 + Please comment in manuscript

- p. 17, Figure 10

 + Text in right half of figure is too small to comfortably read

- p. 21 Figure 13

 + Add title to third column

- p. 23

 + "increase training opportunities in bioimaging"

 + Should, likely, read "increase training opportunities in bioimage analysis"

- p. 35, Figure S1

 + Please create higher quality figure that better supports zooming in

- Suggestion

 + Cite first author's recent paper in F1000R-NEUBIAS on same topic

---

## [Decision Letter · Decision Letter 2]

26 Feb 2021

Dear Tracey,

I've obtained advice from two of the previous reviewers, and on behalf of my colleagues and the Academic Editor, Jason Swedlow, I'm pleased to say that we can in principle offer to publish your Meta-Research Article "Creating Clear and Informative Image-based Figures for Scientific Publications" in PLOS Biology, provided you address any remaining formatting and reporting issues. These will be detailed in an email that will follow this letter and that you will usually receive within 2-3 business days, during which time no action is required from you. Please note that we will not be able to formally accept your manuscript and schedule it for publication until you have made the required changes.

PRESS: We frequently collaborate with press offices. If your institution or institutions have a press office, please notify them about your upcoming paper at this point, to enable them to help maximise its impact. If the press office is planning to promote your findings, we would be grateful if they could coordinate with biologypress@plos.org. If you have not yet opted out of the early version process, we ask that you notify us immediately of any press plans so that we may do so on your behalf.

Thank you again for supporting Open Access publishing. We look forward to publishing your paper in PLOS Biology. 

Best wishes,

Roli

Roland G Roberts, PhD 

Senior Editor 

PLOS Biology

REVIEWERS' COMMENTS:

Reviewer #1:

[identifies herself as Elisabeth M Bik]

I thank the authors for addressing all of the comments raised by the reviewers. I look forward to see this paper published.

Reviewer #2:

[identifies herself as Beth Cimini]

The authors have satisfied my concerns and I can happily recommend this work for publication.